# What are the expectations and experiences of a GMH research programme delivered in Bosnia-Herzegovina, Colombia and Uganda? A prospective longitudinal qualitative study

Vian Rajabzadeh ⬦ , Michael McGrath, Francois van Loggerenberg, Victoria Jane Bird, Stefan Priebe

Unit for Social and Community Psychiatry, WHO Collaborating Centre for Mental Health Services Development, Queen Mary University of London, London, UK

**Correspondence to**
Vian Rajabzadeh;
vian.rajabzadeh@qmul.ac.uk

## ABSTRACT

**Objectives** Global health research collaborations between partners in high-income countries and low-income and middle-income countries (LMICs) aim to generate new evidence, strengthen research capacity, tackle health inequalities and improve outcomes. Previous evaluations of such programmes have identified areas for improvement but consisted only of retrospective experiences. We conducted the first prospective study to assess the initial expectations as well as the final experiences of participants of a global health research programme.

**Design, settings and participants** This study adopted a prospective longitudinal qualitative study, 38 participants of a global mental health research programme with partners in Bosnia-Herzegovina, Colombia, Uganda and the (UK). The interviewees included senior investigators, coordinators and researchers. Framework analysis was used to analyse the data.

**Outcome measures** Participants were interviewed about their initial expectations at the inception of the research programme and their final experiences at the end.

**Results** Many of the original expectations were later reported as met or even exceeded. They included experiences of communication, relationships, developed research expertise, further research opportunities and extending networks. However, other expectations were not met or only partially met, mainly on developing local leadership, strengthening institutional research capacity and opportunities for innovation and for mutual learning. Around equity of partnership and ownership of research the views of participants in the UK tended to be more critical than those of partners in LMICs.

**Conclusions** The findings suggest that global health research programmes can achieve several of their aims, and that partners in LMICs feel equity has been established in the partnership despite the imbalance of the funding arrangement. Aims of global health research projects should have a realistic focus and be proportionate to the parameters of the funding arrangement. More resources and longer time scales may be required to address sustainable structural capacity and long-standing local leadership sufficiently.

## STRENGTHS AND LIMITATIONS OF THIS STUDY

⇒ To our knowledge, this is the first longitudinal qualitative exploration of expectations and experiences of a Global Mental Health collaboration exploring partnership dynamics throughout implementation.
⇒ Social desirability bias may have played a role in the responses of the participants involved in the collaboration.
⇒ The findings are derived from only one research collaboration which is specific to mental health research, therefore one must be cautious when drawing overall conclusions.
⇒ The initial interviews took place at the inception of the group's formation, and it was only the senior investigators who were awarded the funding, therefore, these interviews depict mostly the expectations of these individuals.

## BACKGROUND

Global health research collaborations between organisations in high-income countries (HICs) and low-income and middle-income countries (LMICs) commonly pursue several aims. These can include generating new research evidence, strengthening the research capacity in LMICs, tackling health inequalities across and within countries, and improving the quality and outcomes of healthcare in LMICs. Previous research has developed frameworks to guide such collaborations and identified critical areas for successful, sustainable and equitable cooperation,[1–4] including funding arrangements, rules for authorship of publications, the ownership of research, the contributions of different stakeholders to the research and implementation process, and the building of lasting research capacity in LMICs.

These frameworks were derived from retrospective evaluations of global health research

projects, capturing participants' experiences at a cross-sectional time point, usually after the completion of the project.[5] To our knowledge, there has been no prior research that assessed initial expectations and assumptions about a global health research project of a range of participants in both HICs and LMICs as well as their experiences at the end of the project.[1 6] Prospective longitudinal evaluations can explore how views changed over time and to what extent initial expectations were or were not met. This may help to develop realistic expectations from the beginning and manage expectations during the research to maximise a sense of achievement and reduce potential frustration. Such evaluations should consider the views of different types of participants in the research, that is, senior researchers, managers and researchers who implement the study designs on the ground.

Against this background, we conducted a prospective longitudinal qualitative evaluation of a Global Mental Health Research programme with partners in Bosnia-Herzegovina, Colombia, Uganda and the (UK). The programme focused on developing and testing resource-oriented interventions for people with severe mental illnesses in the three participating LMICs, and this evaluation explored and compared initial expectations and later experiences of the partners.

## METHODS

### Setting

This study evaluated the work of a research programme funded by the National Institute of Health Research (NIHR) in the UK. The NIHR Global Health Research Group on 'Developing Psycho-Social Intervention for Mental Health Care' (GLOBE; August 2017 to March 2022) comprises partners in Sarajevo (Bosnia-Herzegovina), Bogotá (Colombia), Kampala (Uganda) and London (UK), thus including partners in LMICs based in three continents. Further partners in Buenos Aires (Argentina), Karachi (Pakistan) and Lima (Peru) joined the programme later and participated in only very limited activities so that they were not considered in this evaluation. GLOBE aimed to foster relationships between experts in HICs and LMICs and work with local stakeholders to develop and test three resource-oriented interventions for patients with severe mental illnesses. Resource-oriented interventions aim to mobilise and use resources that already exist in communities, families and healthcare systems.

The three interventions were (1) Befriending through volunteers; volunteers regularly met individual patients or small groups of them to provide psychological, social and practical support; (2) Multifamily groups: several patients with family members or friends had regular meetings guided by a mental health professional to exchange experiences and encourage mutual support and learning; and (3) DIALOG+: clinicians and patients used an app-supported intervention to turn routine meetings into therapeutically effective interventions.[7]

The adaptation of each intervention and the design of the study protocols involved local stakeholders—that is, patient groups, clinicians, service managers and policy-makers—to ensure appropriateness and practical relevance for the given context. Each intervention was provided for 6 months with a further 6-month follow-up period. The overall protocol and results of studies have been published elsewhere.[8 9] GLOBE also sought to provide capacity strengthening activities, including regular meetings with senior researchers and research assistants in all LMICs, placements of researchers in the coordinating centre in London, monthly seminars, and extensive training covering the management and analysis of qualitative and quantitative data using relevant software programs.

### Study design and sampling

In a prospective longitudinal qualitative evaluation, we assessed the expectations and experiences of the NIHR Global Health Research Group. Two interviewers conducted two rounds of semistructured one-to-one interviews, between June and December 2017 at the group's inception and between September 2020 and February 2021 towards the end of the programme. The initial interviews were conducted once the programme was initiated.

Participants included senior investigators, project managers and researchers, all involved in the setting up and delivering GLOBE. Most of the interviews on expectations were in-person and took place in each participant's country. Due to the COVID-19 pandemic, interviews on experiences were all conducted online. This study is reported adhering to the guidelines defined by the Standards for Reporting Qualitative Research.[10]

### Data collection

All interviews were conducted in English using semistructured interview guides. The guide for the initial interviews addressed individual expectations, concerns and anticipated challenges of the global mental health research collaboration in GLOBE. The findings informed the guide for later interviews on the experiences (see online supplemental appendix A for the topic guide).

On average, interviews lasted 50 min (range: 30–70 min). All the interviews were recorded on two different devices and transcribed ad verbatim.

### Data analysis

Transcripts were imported into NVivo V.12 and analysed using framework analysis.[11] Initial interviews on expectations were analysed first, and later compared with the interviews on experiences.

Initial interviews on expectations were read several times to ensure familiarity and identify the key themes. Codes were developed and refined until no new aspects were identified and organised into a thematic framework, which the experiences were compared against. Codes for both sets of interviews were first developed by one researcher (VR), and 40% of the transcripts two

**Table 1** Themes and subthemes relating to the key expectations of global collaboration

| Themes | Ensuring group coherence and commitment | Equity in the partnership | Learning and development | Sustainability and impact |
|---|---|---|---|---|
| Subthemes | Clear, regular, transparent communication | Ownership of the research | Developing research expertise | Publications and dissemination |
| | Relationships based on mutual respect | Limitations to partnership in designing the interventions | Opportunity for innovation | New research opportunities and extended networks |
| | Language as a barrier | Coordination and power dynamics | Mutual learning | Investing in local leadership |
| | Commitment to the programme | | Strengthening research capacity | |

secondary reviewers (FVL and MM) conducted independent coding and theme development to ensure trustworthiness of the findings. All researchers determined data saturation during the final stages of the analysis.[12 13] The results were regularly discussed in the team of authors who were all involved in global health research, had grown up on different continents, had different clinical and nonclinical backgrounds and were at different stages of their career, and also in the wider multidisciplinary research group of the Unit for Social and Community Psychiatry. Reflexivity was continuous, from the stages of data collection to manuscript development.[14]

## Patient and public involvement

It was not appropriate to involve patients or the public in the design, or conduct, or reporting, or dissemination plans of our research.

## RESULTS
### Sample characteristics

Thirty-eight participants were interviewed (for professional characteristics see online supplemental appendix B). Initial expectations were assessed in 19 and experiences in 30 interviews. Thus, there were 49 interviews in total, with 11 participants being interviewed about both initial expectations and later experiences. Three participants who for different reasons left the programme midway were also interviewed about their experiences to include potentially more negative views of participants who discontinued their involvement.

## Overall framework

The overall framework, presenting the main expectations derived from the interviews, is shown in table 1.

Table 2 shows how the experiences fit into the following categories: (1) expectations met; (2) expectations exceeded; (3) expectations partially met and (4) expectations not met. The results section is structured using this categorisation and additional quotes to illustrate each category are provided in text boxes (online supplemental tables 1–4).

## Expectations met
### Clear, regular, transparent communication

The respondents hoped for clear, ongoing communication among the wider research group to ensure a joint commitment to the programme.

> Communication is so important to make sure there are no misunderstandings and people remain committed to the programme. (R-16 Ugandan Senior Investigator expectations)

> I think productive communication needs regular communication. (R-32 UK Senior investigator expectations)

**Table 2** Expectations met, exceeded or partially met

| Expectations met | Expectations exceeded | Expectations partially met | Expectations not met |
|---|---|---|---|
| Clear, regular, transparent communication | Commitment to the research | Ownership of the research | Opportunity for innovation |
| Relationships based on mutual respect and trust | New research opportunities and extended networks | Limitations to partnership in designing the interventions | Mutual learning |
| Language as a barrier | | Coordination and power dynamics | |
| Developing research expertise | | Investing in local leadership | |
| Publications and dissemination | | Strengthening research capacity | |

Transparency was hoped for to ensure a shared understanding of all processes within the project, particularly for those who had worked in previous collaborative projects where they reported that important processes were kept hidden.

> There were many other projects also regional I was involved in. […] All these projects were done behind closed doors. […] And this happened as I said is the general culture in our country. […] I'm not saying that everyone should be involved, but some transparency should be there. (R-02 Bosnian Researcher's expectations)

All participants felt that clear communication was sustained throughout. The regular meetings enabled a collective awareness throughout the programme, which many acknowledged as valuable.

> So I think the facilitators of the project have maintained open communication lines, in that anytime you have a challenge, you can reach out. (R-20 Ugandan Coordination/management experiences)

> And when we hear about the work in different places, I think it's important for the group's creativity. (R-07 Colombian Coordination/management experiences)

LMIC partners felt that being involved in the initial stages of setting up the studies, ensuring all were copied in on correspondence relevant to them, and an explicit authorship policy contributed to the transparency experienced.

> I would say yes especially with the UK team and our local team and the PI, there was transparency […] You were present at our meetings with the finance team, with the admin team. So we always knew what was happening. (R-04 Bosnian Researcher experiences)

### Relationships based on mutual respect and trust

Given that participants would be working across different contexts, it was expected that relationships convey mutual respect, display cultural sensitivity and accommodate different working styles.

> It's about the people, the relationship that you develop with people once it is solid, then you can always move forward. (R-22 Ugandan Senior Investigator expectations)

Many participants experienced mutual respect in relationships, to the extent of facilitating new research opportunities. One researcher reflected on their role in developing new research studies as an extension to the original GLOBE study:

> My opinion was respected. My ideas were respected. And the idea to research DIALOG+ in primary health care was mine. So yes, I feel quite respected. (R-01 Bosnian Senior Investigator experiences)

### Language as a barrier

Language was also identified as a potential concern in the context of working across multiple countries, especially with the partner groups being expected to understand and relay complex information to the rest of the group when needed, and articulating ideas during the teaching weeks.

> But really understanding takes time. So that's one barrier. Language is another barrier. Communication and everybody because communication doesn't work smoothly. (R-06 Colombian Senior Investigator expectations)

Despite initial concerns, individuals did observe how language impacted on the capacity to work collaboratively and communicate effectively across the countries.

> The other thing is that language is a huge barrier. So, when you ask about mutual learning, about collaboration, they face a barrier in the language. (R-06 Colombian Senior Investigator experiences)

### Developing research expertise

Individuals expected to develop their understanding of research methods and designs and learn how to conduct high-quality research.

> And this is also rewarding because we'll develop methodological skills and research-related skills like writing papers or projects or applying for funds (R-01 Bosnian Senior Investigator expectations)

Many respondents outlined the specific research skills they gained from the collaboration, including defining and standardising procedures to ensure consistency and reduce errors when implementing specific tasks.

> I learned about the protocols, and how we make protocols for everything, and present that information to the sites. I didn't do that kind of work before, and I think it was very useful …(R-07 Colombian Coordination/management experiences)

### Publications and dissemination

Publications were considered a vital output of the research collaboration, allowing researchers to exhibit competency to the research community, and support career development.

> So I think that comes from other research because they are very important for the careers, for us recently publishing has become more important. (R-06 Colombian Senior Investigator expectations)

Experiences of the publication process were perceived as positive. Early-career researchers from LMIC partners were given the opportunity to be the lead author on papers and contribute contextual insight gained from working directly with the intervention and its recipients.

We were given an opportunity to write […] do the literature review, and be genuine with what has been happening in the hosting community. (R-27 Ugandan Researcher experiences)

## Expectations exceeded
### Commitment to the research

Since not all researcher assistant had been recruited when senior researchers in LMICs were interviewed about their expectations, some expressed doubts about whether research assistants would remain committed to the research programme.

I hope I make the right choice for the research assistants […] Because if I train someone to deliver the interventions, and they decide to leave after three, six months, it will be necessary to train another. (R-01 Bosnian Senior Investigator expectations)

Yet when discussing the commitment of the group's members, including the researcher assistants, many participants remarked on their enthusiasm and dedication, suggesting that the experiences exceeded initial expectations.

I think what I really enjoyed about working on the project was the people. So, everyone on the teams were very nice people to work with but also very engaged, interested, enthusiastic about the work and very hard working. (R-36 UK Coordination/management experiences)

### New research opportunities and extended networks

Respondents anticipated that participating in the GLOBE programme would lead to further research opportunities.

Then research opportunities will come out of this, depending on how much effort are you putting in. (R-24 Ugandan Coordination/management expectations)

Indeed, several new research projects emerged from the GLOBE programme that received competitive funding, indicating that expectations were exceeded. One study, led by the Ugandan research group, explored patient support during consultations:

The idea for the first proposal came from the Uganda team, but was co- developed together with the UK team. The things we wanted to appreciate were the reasons for patients coming back for review and who is supporting them in doing this (R-17 Ugandan Senior Investigator experiences)

As a result of additional funding to the site in Colombia the network expanded in Latin America.

We are planning another network with two countries of Latin America … we could help both groups, groups that are intermediate like ours and groups that are beginning. (R-05 Colombian Senior Investigator experiences)

## Expectations partially met
### Ownership of the research

Partners expressed a desire for autonomy and ownership when describing their ideal collaborations, especially being responsible for their studies.

The best collaborations I've had are when they let me be their driver because I know the system […] but they feel like they should control what's going on locally and usually makes you feel disempowered. (R-17 Ugandan Senior Investigator expectations)

LMIC partners perceived the collaboration as meeting their expectations for acquiring ownership of their studies. In contrast, UK participants believed this difficult to realise when the whole programme is being funded by one country.

So, to me whenever there are institutions from other countries, as long as I have ownership, I tend to like it better. You know, it's better organised, you know, some things to learn from them because they're from different cities. (R-21 Ugandan Coordination/management experiences)

### Limitations to partnership in designing the interventions

Regarding partners' contributions, a UK senior researcher emphasised that the collaboration would be a space where every member could contribute their perspectives and input. Partners' expected their knowledge of the local context and health systems to help adapt the interventions and foresee any likely challenges.

I think Queen Mary already has a protocol somewhere, but I think we're going to have to put in the nitty-gritty details for the process of the adaptation (R-23 Ugandan senior investigator expectations)

Although the collaboration created space to share and exchange input, when asked about contributions made towards adjusting the interventions, a Colombian senior investigator commented:

The research designs and many of the main components come from the UK, the role of Colombia or other countries is limited because the money and the resources are not ours. So that means that the possibility of really making changes or deciding many things about the project is limited. (R-06 Colombian Senior Investigator experiences)

Participants in the UK echoed this opinion when asked about how the partner's contributed to this process.

If they needed things changed, they did put their case forward. But because they were all interventions that were developed in the UK, I suppose they went with the flow for a lot of it, just to test things out. (R-37 UK Coordination/management experiences)

## Coordination and power dynamics

The UK group expected to provide administrative and research support during the programme's roll-out, while anticipating the challenges around ensuring their involvement was not too prescriptive. There were concerns about the uneven distribution of power:

> Rather than having a partnership of four equal sites, it still looks like you have one side that is partnering down on the three other sites and setting the agenda. I know this is where the research expertise is. (R-35 UK Senior Investigator expectations)

The need to meet the grant requirements imposed a way of coordinating the group in a more prescriptive manner than anticipated and influenced the power dynamic within the collaboration.

> I think we're quite restricted by the actual mechanisms of the grant and things such as the fact that the contracts must be issued through Queen Mary […] it all rests with the lead organisation [the UK] (R-33, UK Senior Investigator experiences)

The LMIC partners did not comment on the presence of a power dynamic, but rather around the consistent and constructive support they had received.

> I would like to say that the UK team was immensely supportive. At times I felt like we were pestering them, they had this infinite patience for us and our constant questions. So I think none of this would have gone as quickly and well as it did if we weren't sort of supervised by the UK team (R-04 Bosnian Researcher experiences)

## Investing in local leadership

Investing in and developing in local leadership was recognised as essential for working toward the sustainability of the research groups and a key expectation of the programme.

> I would have the opportunity to employ three young researchers. The project will employ them, we will have them in the department, and they will simultaneously be acquiring research skills in collaborations with Queen Mary and Uganda and Colombia. And they will remain an asset to the department where I work (R-01 Bosnian Senior Investigator expectations)

Participants felt that the grant lacked the resources to make the infrastructural changes needed to establish academic posts.

> So I think that that role that it had being able to help other people, to develop their career has been fulfilled with the limitation of the structure of any faculty that is flexible, but it's not entirely flexible to changes. (R-06 Colombian Senior Investigator experiences)

## Strengthening research capacity

Building on and strengthening research capacity was a significant expectation, with one respondent viewing it as a central part of the collaboration.

> We don't have the capacity to do some things. For example, we don't have capacity to successfully submit a Wellcome Trust grant and win it without help. So, for selfish purposes, we need to build our capacity. (R-22 Ugandan Senior Investigator expectations)

One participant perceived capacity building as developing skills at the individual level to deliver the current programme and achieve it.

> There was need for capacity building for the members on the team at different stages of the study […] we needed to train the researchers in REDCap, data entry, collecting data for qualitative interviews, reviewing transcripts, all that was part of the capacity building that has been emphasised through the study (R-20 Ugandan Coordinator experiences).

Although the pandemic hindered some aspects of capacity strengthening, a UK respondent considered the programme's efforts inadequate overall.

> I'm not so sure. It was difficult. Yes, of course, we build up research capacity a bit, but if the whole group stopped tomorrow, we wouldn't leave long-term, highly functioning research groups behind. (R-32 UK Senior Investigator experiences)

## Expectations not met
### Opportunity for innovation

There was an expectation that working in resource-limited contexts and collaborating with international experts would lead to new ideas and interventions, given that constraints can lead to innovation.

> So, looking at different cultures and seeing how distress is dealt with around the world can be one way to get new perspectives that could lead to real innovation rather than just I'm going to tweak this intervention slightly or I'm going to try this intervention with a different population (R-34 UK Senior Investigator expectations)

The LMIC partners expected to learn more about psychosocial interventions and new treatment approaches that are not common in LMIC contexts. The experience of delivering the interventions fulfilled the expectations of learning about novel, low-cost interventions.

> So I think this is very important because it shows us new opportunities and new ways to help people with a mental concern. […] And it's very cheap. So I think is it is a new way that we have not explored yet enough. I also saw these interventions reduce stigma which is very high in Colombia (R-13 Colombian Researcher experiences)

But the expectation of working collectively to generate new ideas for interventions in the context of was ultimately not met.

> Maybe the thing that we have still need to do is how to develop research ideas collectively […].I would like to learn how to work with a group and think together to develop new research ideas. (R-06 Colombian Senior Investigator experiences)

### Mutual learning
In the expectation interviews, a key motivation for international collaboration was the strong desire to work collaboratively with a diverse group of researchers and promote cross-cultural discussion and learning.

> Mutual learning means sharing experience and discussing different points of views. (R-02 Bosnian Researcher expectations)

A UK senior investigator expressed doubts about the arrangements established to encourage mutual learning, such as the teaching weeks and seminars being hosted in the UK.

> My understanding is that lots of the sharing and learning is going to be done in Britain and I suppose you're out of your comfort zone in somebody else's country and you don't own it as much. (R-35 UK Senior Investigator expectations)

While partner perspectives demonstrated the development of research expertise, learning on the UK side was less apparent. Although the UK team did not necessarily acquire research skills, one UK investigator acknowledged:

> One of the things I've personally learned from Uganda approach is how better to include different stakeholders. They're very good at hearing multiple voices in the research and to deal with that in a sensitive way that everybody feels heard (R-33 UK Co-investigator experiences)

Generally, some interviewees perceived mutual learning to be even less evident among the partner groups, perhaps due to the lack of interaction between them.

> There should be intercommunication between the different players, a lot of communication with the other institutions as opposed to the communication being only between, Uganda and Queen Mary (R-16 Ugandan Senior Investigator experiences).

## DISCUSSION
### Main findings
The findings indicate that most expectations were either partially met, met or exceeded, and there were hardly any unexpected challenges. Expectations were met concerning good and open communication, collegiate and trustful personal relationships, developed individual research expertise, further research opportunities and extending professional networks. However, other expectations were not met or only partially met. They were about developing local research leadership, strengthening institutional research capacity and opportunities for innovation and for mutual learning. Around equity of partnership and ownership of research the views of participants in the UK tended to be more critical than those of partners in LMICs.

### Interpretations and comparisons with the existing literature
Most of the identified expectations and experiences address aspects previously raised in the literature, although not necessarily in the nuanced way as in this evaluation. Many initial expectations were met or even exceeded, and the general tone of experiences was positive. Central to the positive experiences appear good personal relationships with open, regular and inclusive communication, mutual trust and respect for everyone involved. Still, some expectations remained not or only partially fulfilled.

The latter included the hope for mutual learning, both between LMICs and HICs and among the partners in LMICS themselves. While partners in LMICs were satisfied with what they had learnt through the research activities guided by the centre in an HIC, they felt they had learnt rather little from each other.[8 9] Having a site in an HIC as the coordinating centre, being located on different continents, working in very different contexts, establishing relationships with new partners, and having different mother tongues may have hindered direct exchange and interaction among the LMIC partners. Subsequently, the main relationships from the collaboration that led to further successful grant applications were bilateral between the centre in the UK and partners in the different LMICs. Explicitly identifying what all partners may learn from each other could be discussed throughout collaborations to foster mutual learning.[1]

Another related disappointment was the limited scope for developing innovative ideas. Much time was dedicated to establishing relationships and delivering what the group had promised to deliver in the grant application, which may have limited the options for creative and innovative thinking.[15–17] At the same time, the feeling of a lack of innovation might be a wider phenomenon in mental health research and be only reflected in global health research rather than specifically arising in it.[18]

Achieving equitable relationships is a crucial goal for many global health research collaborations.[4] The literature highlights how the dynamic imposed by Western funding structures can impact the equality of a partnership, especially with the obligations of meeting the funding expectations.[2] Similar concerns were initially expressed by participants in this evaluation, more so from UK participants than LMICs. Overall, participants in the UK remained sceptical about a true and equal partnership until the end. In contrast, most participants in LMICs felt their initial hopes for equity among partners had actually

been met and this occurred despite the restrictions and potentially paternalistic nature of funding channelled by an HIC that all partners had been aware of from the beginning.[2 4 17] Again, communication and relationships appear central to this.

Similarly to the positive experience of equity, partners in LMICs also perceived expectations of capacity building as fulfilled, a view that participants in the UK did not share. In the literature, there are different understandings of what capacity building entails; some view it as training related to the current research project, whereas others view it as enhancing infrastructural support.[4] Addressing both individual and organisational aspects and balancing the development of project-specific and general skills are required to establish sustainable research groups in LMICs.[19 20] All participants agreed that the research expertise of various individuals in each country had markedly improved, also benefiting from individual mentoring and longer spells of some researchers at the co-ordinating centre in London.[21 22] Yet, there were doubts as to whether the progress of individuals would lead to a sustained increased research capacity on an institutional level when there was no infrastructure for research careers and respective funding. Related to this, participants in all countries considered that the efforts to invest in local leadership were beyond the research programme's resources and that more resources and particularly longer time-scales were needed to ensure the continuity of research posts and, subsequently, research infrastructure.[23]

### Implications for research and practice

The study evaluated a global health research programme that was relatively successful in terms of conventional academic outcome criteria: despite the unforeseen complications through the pandemic all trials were completed as planned in the protocol, the tested interventions were shown to be feasible and beneficial, and various results were published in peer-reviewed journals. Meeting or even exceeding these criteria was mentioned in the reported experiences, although it did not dominate them. Yet, meeting the conventional aims of research projects may still have been the basis for the generally positive perception of the overall research programme. Experiences were favourable on a number of aspects of the research programme and they underline the importance of investing enough time and energy into establishing transparent communication and trustful relationships from the very beginning.

With respect to the areas of disappointment—developing local leadership, strengthening institutional research capacity, and opportunities for innovation and for mutual learning—the question arises as to whether research collaborations can and should put more emphasis on these aspects from the outset or whether achieving all the aims of global health research within one programme is unrealistic.

Expectations relating to building institutional research capacity and investing in early-career researchers need to be realistic and proportionate to the amount of funding and time available within a single programme. While more resources and a longer time scale are likely to help strengthen institutional research capacity, changes in the options and arrangements for academic funding in LMICs may also be required so that there are realistic career paths with sufficiently paid long-term positions available to early-career researchers.

### Strengths and limitations

To our knowledge, this is the first prospective longitudinal qualitative evaluation of a global health research collaboration, assessing expectations and corresponding experiences. Also, the study includes perspectives from a multidisciplinary research group and participants at a different career stage across three continents.

The study also has several limitations. First, a social-desirability bias might have influenced participants' responses. Second, the study assessed expectations and experiences of only one research collaboration, potentially making the findings specific to research in mental health and this programme's context. Third, we assessed only the view of the researchers in the group, not of other stakeholders or the funding body. Finally, one can only speculate whether the collaboration might have been different without the restrictions of the pandemic.

### CONCLUSION

The evaluation suggests that many initial expectations and hopes for the outcomes of a global health research programme can be met. Establishing good communication and mutually trustful relationships are central, yet not sufficient to ensure that all initial aims are finally achieved. Participants in HICs were more sceptical in their eventual appraisal than those in LMICs. Evaluations of other global health research programmes should explore whether this reflects a general trend. In any case, it shows that the views of different participants can vary significantly and that all need to be considered in an evaluation of a global health research project.

The funding imbalance in global health research is difficult to change, but this study shows that nevertheless researchers in LMICs can feel equity and fairness in partnerships. At the same time, it may be helpful to identify the expectations of all participants at the outset and monitor progress against them, not only against the milestones defined in the grant application.

Funding bodies on global health research may want to consider whether it is helpful to define a wide range of aims, some of which may be unrealistic to achieve in one single programme. Finally, higher-level agreements with established or potential research institutions in LMICs may be required to secure options for long-term research careers and strengthen sustainable research capacity.

**Correction notice** This article has been corrected since it was first published. Name of the author Francois van Loggerenberg is now correct.

**Contributors** VR conceptualised and is the guarantor of this study. VR, MM and FVL contributed to the data analysis. All authors (VR, MM, FVL, VJB and SP) contributed to the data interpretation and drafting of the manuscript. All authors (VR, MM, FVL, VJB and SP) approved the final manuscript and were responsible for the decision to submit it for publication.

**Funding** This research was funded by the National Institute for Health Research (NIHR) (Global Health Research Group on Developing Psychosocial Interventions for Mental Health Care, project reference 16/137/97) using UK aid from the UK Government to support global health research.

**Disclaimer** The views expressed in this publication are those of the authors and not necessarily those of the NIHR or the UK Department of Health and Social Care.

**Competing interests** None declared.

**Patient and public involvement** Patients and/or the public were not involved in the design, or conduct, or reporting, or dissemination plans of this research.

**Patient consent for publication** Not applicable.

**Ethics approval** This study involves human participants and was approved by Queen Mary, University of London, QMREC2047a. Participants gave informed consent to participate in the study before taking part.

**Provenance and peer review** Not commissioned; externally peer reviewed.

**Data availability statement** No data are available.

**ORCID iD**
Vian Rajabzadeh http://orcid.org/0000-0002-1123-9805

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
