## [Reviewer comments · BMJ Open]

ARTICLE DETAILS

TITLE (PROVISIONAL)	What are the expectations and experiences of a GMH research programme delivered in Bosnia-Herzegovina, Colombia and Uganda? A prospective longitudinal qualitative study
AUTHORS	Rajabzadeh, Vian; McGrath, Michael; van Loggerenberg, Francois; Bird, Victoria; Priebe, Stefan

VERSION 1 – REVIEW

REVIEWER	Jenkins, Rachel King's College London Institute of Psychiatry Psychology and Neuroscience
REVIEW RETURNED	04-Jan-2022

GENERAL COMMENTS	This is a most interesting prospective qualitative study of researcher expectations and experiences in a multicountry research programme across high, middle and low income countries. One of the findings was that LMIC researchers did not have the opportunity to work collectively to generate new ideas for research, and I assume this arose because they had not been equal partners in the development of the research programme that was ultimately funded. Perhaps the discussion section could clarify how the programme was originally developed and whether the baseline interviews took place before notification of the grant award or afterwards, once the programme was initiated.
---

REVIEWER	Hook, Kimberly Boston Medical Center, Psychiatry
REVIEW RETURNED	02-Apr-2022

GENERAL COMMENTS	Thank you for the opportunity to review this manuscript. I found the topic to be compelling, and I appreciate the authors' attention to this topic. I have no substantial edits, only 2 comments. First, moving the strengths/limitations section of the manuscript towards the end of the discussion may improve the article flow. Second, the authors might consider mentioning saturation and/or how they determined they collected enough data to cease data collection. Very well done, very informative, and very useful.
---

VERSION 1 – AUTHOR RESPONSE

Reviewer 1

This is a most interesting prospective qualitative study of researcher expectations and experiences in a multicountry research programme across high-, middle- and low-income countries. One of the findings was that LMIC researchers did not have the opportunity to work collectively to generate new

ideas for research, and I assume this arose because they had not been equal partners in the development of the research programme that was ultimately funded. Perhaps the discussion section could clarify how the programme was originally developed and whether the baseline interviews took place before notification of the grant award or afterwards once the programme was initiated.

Author response: Many thanks for your insightful comment. In response to this comment, I have included a sentence clarifying that the baseline interviews took place after the programme was initiated.

The revised text reads as follows on page 4 of the revised manuscript:

“The initial interviews were conducted once the programme was initiated.”

Reviewer 2

Thank you for the opportunity to review this manuscript. I found the topic to be compelling, and I appreciate the authors’ attention to this topic. I have no substantial edits, only 2 comments. First, moving the strengths/limitations section of the manuscript towards the end of the discussion may improve the article flow. Second, the authors might consider mentioning saturation and/or how they determined they collected enough data to cease data collection. Very well done, very informative, and very useful.

Author response: Thank you for your comment. I have moved the strengths/limitations section to feature before the conclusion. I have acknowledged how we reached data saturation in the methods section.

The revised text reads as follows on pages 4 & 15 of the revised manuscript:

“All researchers determined data saturation during the final stages of the analysis (12,13).”